# Mechanoradicals in tensed tendon collagen as a source of oxidative stress

Christopher Zapp [1,2,12], Agnieszka Obarska-Kosinska[1,3,12], Benedikt Rennekamp[1,2], Markus Kurth[1], David M. Hudson[4], Davide Mercadante[5], Uladzimir Barayeu[6,7], Tobias P. Dick[7], Vasyl Denysenkov[8], Thomas Prisner[8], Marina Bennati[9], Csaba Daday[1,10], Reinhard Kappl[11] & Frauke Gräter [1,10 ✉]

As established nearly a century ago, mechanoradicals originate from homolytic bond scission in polymers. The existence, nature and biological relevance of mechanoradicals in proteins, instead, are unknown. We here show that mechanical stress on collagen produces radicals and subsequently reactive oxygen species, essential biological signaling molecules. Electron-paramagnetic resonance (EPR) spectroscopy of stretched rat tail tendon, atomistic molecular dynamics simulations and quantum-chemical calculations show that the radicals form by bond scission in the direct vicinity of crosslinks in collagen. Radicals migrate to adjacent clusters of aromatic residues and stabilize on oxidized tyrosyl radicals, giving rise to a distinct EPR spectrum consistent with a stable dihydroxyphenylalanine (DOPA) radical. The protein mechanoradicals, as a yet undiscovered source of oxidative stress, finally convert into hydrogen peroxide. Our study suggests collagen I to have evolved as a radical sponge against mechano-oxidative damage and proposes a mechanism for exercise-induced oxidative stress and redox-mediated pathophysiological processes.

[1] Heidelberg Institute for Theoretical Studies, Schloss-Wolfsbrunnenweg 35, 69118 Heidelberg, Germany. [2] Institute for Theoretical Physics, Heidelberg University, Philosophenweg 16, 69120 Heidelberg, Germany. [3] Hamburg Unit c/o DESY, European Molecular Biology Laboratory, Notkestrasse 85, 22607 Hamburg, Germany. [4] Department of Orthopaedics and Sports Medicine, University of Washington, Seattle, WA 98195, USA. [5] Biochemical Institute, University of Zuerich, Winterthurerstr. 190, 8057 Zuerich, Switzerland. [6] Faculty of Biosciences, Heidelberg University, Im Neuenheimer Feld 234, 69120 Heidelberg, Germany. [7] Division of Redox Regulation, DKFZ-ZMBH Alliance, German Cancer Research Center (DKFZ), Im Neuenheimer Feld 280, 69120 Heidelberg, Germany. [8] Institute of Physical and Theoretical Chemistry, Goethe University Frankfurt, Max-von-Laue-Str. 7, 60438 Frankfurt am Main, Germany. [9] Max Planck Institute for Biophysical Chemistry, Am Fassberg 11, 37077 Göttingen, Germany. [10] Interdisciplinary Center for Scientific Computing, Heidelberg University, INF 205, 69120 Heidelberg, Germany. [11] Institute for Biophysics, Saarland University Medical Center, CIPMM Geb. 48, 66421 Homburg/Saar, Germany. [12]These authors contributed equally: Christopher Zapp, Agnieszka Obarska-Kosinska. ✉email: frauke.graeter@h-its.org

Polymers subjected to mechanical stress—be it a shoe sole or rubber band—generate mechanoradicals by undergoing homolytic bond scission[1,2]. Radicals form in polymers even in presence of water, and then readily convert into reactive oxygen species (ROS)[3,4]. A prime candidate for mechanoradical formation in biological polymers is collagen. As the basic material of tendons, cartilage, ligaments, and other connective tissues, collagen is under perpetual mechanical load, and provides structural and mechanical stability to virtually all human tissues. The viscoelastic properties of collagen have been studied in depth[5,6], and have been ascribed to a hierarchical deformation mechanism, involving straightening and shearing between and within fibrils and triple helices, as well as triple helix unwinding and eventually covalent bond rupture[7–14]. The only evidence for radicaloid bond rupture in biomaterials stems from electron-paramagnetic resonance (EPR) experiments of cut fingernails and milled bone[15,16]. If and how mechanoradicals form and function in protein materials under physiological, that is, subfailure levels of loading are fully unknown.

Here, we performed tensile tests of fascicles from rat tail tendon and detected radicals originating from mechanical bond scission, using EPR spectroscopy. We built an atomistic model of collagen I fibril and identified the regions around crosslinks to carry maximal stresses in molecular dynamics (MD) pulling simulations. Our joint simulations and experiments show that radicals from primary irreversible bond scission in these crosslink regions migrate to dihydroxyphenylalanine (DOPA), which form by posttranslational modifications from phenylalanine and tyrosine residues. These aromatic residues cluster in evolutionarily highly conserved regions. We suggest these clusters of conserved aromatic residues as possible sponges for mechanoradicals, preventing oxidative damage to the tissue. Furthermore, we showed with light absorbance that the DOPA radicals were finally converted into hydrogen peroxide in the presence of water, putting forward a role of collagen in the conversion of mechanical into oxidative stress in connective tissues.

## Results

**Collagen forms radicals under load**. To detect tension-induced radical formation in collagen, we subjected fascicles dissected from rat tail tendon to constant forces of 5, 10, 15, and then 20 N, each for 1000 s (Supplementary Fig. 1a). These forces correspond to stresses in the 2–40 MPa range, depending on the exact fascicle diameter (see "Methods" section). They thus fall into the regime of physiological stresses of tendons, which can exceed ten times the body weight (~90 MPa), and also lie well below the rupture stress of fascicles[17]. In this force regime, we recovered the known force-extension behavior of collagen fascicles (Supplementary Fig. 1b)[5]. Before and in between each constant loading period, we directly measured the presence of radicals in the fascicle by continuous wave (cw) X-band EPR spectroscopy, which is an established method to probe polymer mechanoradicals[18,19]. With increasing constant force, we observed a significant increase in the EPR signal, which was defined as the difference between the minimal and maximal intensity (Fig. 1a, b). We could reproduce the increase in radicals with the load for different fascicles from the same rat, as well as for fascicles from different rats. The increase in the EPR signal was also observed when extending the duration of force application while keeping the force constant (Supplementary Fig. 1c) and also under cyclic stretching (Supplementary Fig. 1d). We note that collagen tendons show already a small signal (Fig. 1a, prior to loading), putatively due to the minor load present during extraction (see "Methods" section). Marino and Becker[20] describe a similar EPR signal (g-factor ≈ 2.007, width ≈ 10 G) originating from non-stressed tendons,

without giving any interpretation. To directly monitor radical production during loading, we devised an EPR setup that allowed to mount the fascicle within the EPR cavity, while being subjected to 3.43 N of load (Fig. 1c). Again, within the timescale of tens of minutes, we observed a steady increase in the amount of free radicals during loading. Radicals remained stable over many hours at the measurement conditions (room temperature (RT), 25% relative humidity). We validated that covalent bond rupture within collagen alpha chains indeed occurs at the external loads applied in the EPR experiments by SDS–PAGE and densitometry analysis (Supplementary Fig. 1e, f). The degradation products were identified by mass spectrometry as $\alpha$-chain fragments of collagen I, strongly suggesting that the observed radicals originate from force-induced bond cleavage.

**Bond rupture is preferred in the vicinity of crosslinks**. To answer where covalent bonds rupture and radicals accumulate in collagen materials under subfailure load, we built an atomistic model of a collagen fibril using an integrated structural modeling approach based on a low-resolution fibril structure[21] and high-resolution collagen-like peptide structures. The resulting model of 67 nm length spans one overlap and one gap region[21,22] of a bundle of 37 aligned collagen I triple helices (Fig. 2a). It contains 12 hydroxylysino-keto-norleucine (HLKNL) crosslinks formed by specific lysine/hydroxylysine side chains[23] (Fig. 2b). We applied a constant stretching force of 1 nN per chain to the fibril model in MD simulations. We then monitored the distribution of the external force through the fibril structure by force distribution analysis[24]. As expected, the external force propagates along the triple helices and passes over through the crosslinks to adjacent helices (Fig. 2a). Telopeptides show minimal forces in their backbone bonds as they lie outside of these force propagation pathways. The highest forces concentrate in the N-terminal crosslinks and the backbone of approximately ten adjacent residues along the force propagation path (Fig. 2b, c), suggesting bonds in the ~4 nm wide cross sections around these crosslinks to be primary regions of bond rupture and radical formation. Indeed, bond scission events primarily, albeit not exclusively, occur in this cross section according to kinetic Monte Carlo/MD simulations of the same collagen system[25]. C-crosslinks show a less pronounced stress concentration and distribute stress more widely into the gap region (Fig. 2b, c). Under heterogeneous loading of the collagen triple helices, the dominant loading scenario in a not fully flawless natural tissue, we detected maximal stresses in both N- and C-terminal crosslinks and in their direct vicinity (Supplementary Fig. 2b). Taken together, crosslinks and the backbone in their direct vicinity are most stressed. Therein, the candidates for bond scission are the single bonds, which are of either C–C or C–N type. These two bond types exhibit dissociation energies in the same range, 352–377 kJ/mol[26]. CASPT2 calculations of representative crosslink and backbone fragments yielded 352–355 kJ/mol, and confirmed that both C–C and C–N bonds are candidates for mechanoradical formation by homolytic bond scission (Supplementary Fig. 2e).

**Mechanoradicals are stabilized at aromatic residues**. The X-band cw-EPR signal shown in Fig. 1a is not in line with primary radicals generated from C–N or C–C bond scission, as hyperfine couplings from hydrogen or nitrogen are not resolved, suggesting rapid radical migration through electron transfer reactions. Mechanically stressing collagen at 77 K to prevent radical migration yields a distinctly different EPR signal (a typical peroxy signal with a minor contribution from a methylene signal, suggesting oxygen addition to primary radicals at 77 K, Supplementary Fig. 3b, c). We could not detect this signal at RT,

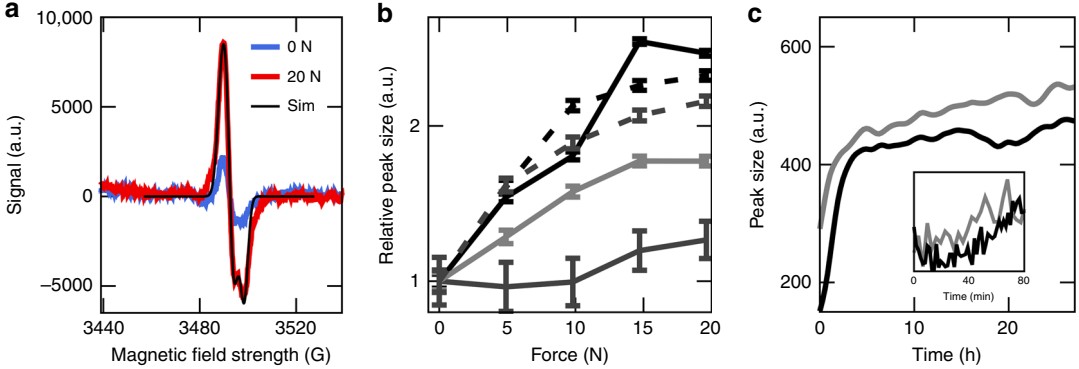

**Fig. 1 Formation of radicals in collagen tendon due to external force. a** Cw-EPR signal (X-band) against the magnetic field strength measured at RT. Blue: unstressed fascicle, red: pulled fascicle, and black: simulation of pulled fascicle signal. **b** EPR signal size (difference between minimal and maximal intensity) relative to the initial signal before loading. Five independent fascicle pulling experiments in different gray tones, continuous and dashed lines for samples pulled at 25% and 45% relative humidity, respectively. Force was removed before the EPR measurement. The EPR signal is collected directly after pulling tendons with the forces shown on the x-axis. Errors show standard deviations calculated from 20 EPR sweeps. **c** Spline-smoothed EPR signal size of two different tendons (black and gray) being pulled directly inside the EPR cavity, while measuring the signal for 27 h, i.e., force was applied during the EPR measurement. The inset shows a zoom for the first 80 min of the non-smoothed EPR signal size.

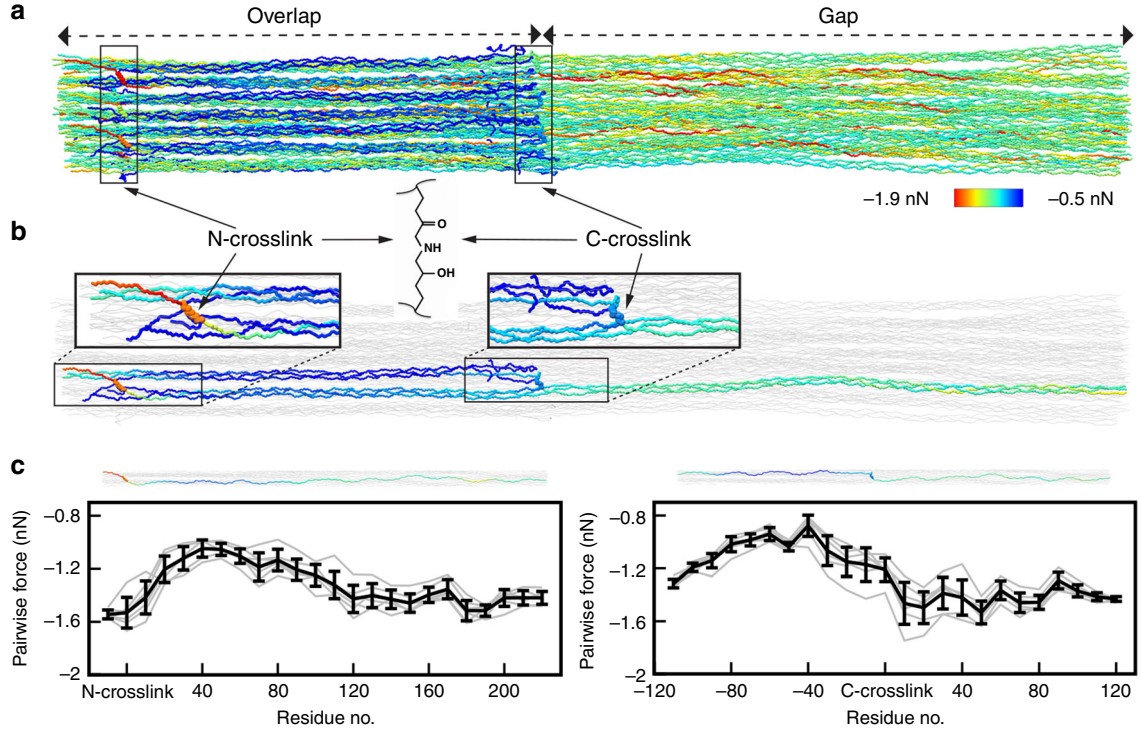

**Fig. 2 Force concentrates around crosslinks in tensed collagen. a** Snapshot of an MD simulation of the crosslinked collagen I model fibril under 1 nN of external force per chain, colored according to the distribution of the external force through the fibril (blue = low force, red = high force). **b** Example pair of overlapping triple helices connected by crosslinks from the snapshot shown in **a** depicted separately to better visualize forces around the crosslinks. Crosslinks are represented as spheres, the remaining collagen chains as gray ribbons. **c** Pairwise forces averaged over ten consecutive residues along the collagen chains connected by crosslinks (left for N-crosslinks, right for C-crosslinks, with one of the chains shown above the plot aligned to the x-axis) for all six pairs of overlapping triple helices (gray) and average with standard errors over the six pairs (black).

suggesting peroxyradicals to be either not present or short-lived intermediates at this temperature. To identify the radical dominant at RT, we performed high-frequency G-band (180 GHz) EPR experiments (Fig. 3a). The obtained signal features a very characteristic axial g-tensor, with g-factors typical for phenoxy-type radicals (Supplementary Tables 1 and 2), and no resolved hyperfine coupling, consistent with the X-band EPR spectrum (Fig. 1a). However, to our knowledge, such an axially symmetric tensor has never been observed in tyrosyl radicals and therefore

strongly suggests the formation of another type of phenoxy radical here. An axial g-tensor has been observed by some of us in a DOPA radical[27], an extraordinarily stable tyrosyl radical with an additional hydroxyl group (Fig. 3b). Interestingly, recent reports on stable DOPA radicals in ribonucleotide reductase reported a rhombic g-tensor[28,29]. We clarified with DFT calculations that the axial g-tensor and the strongly diminished $C_\beta$-hydrogen coupling ( <3.6 G, see Supplementary Table 1) in stressed collagen originate from a deprotonated DOPA

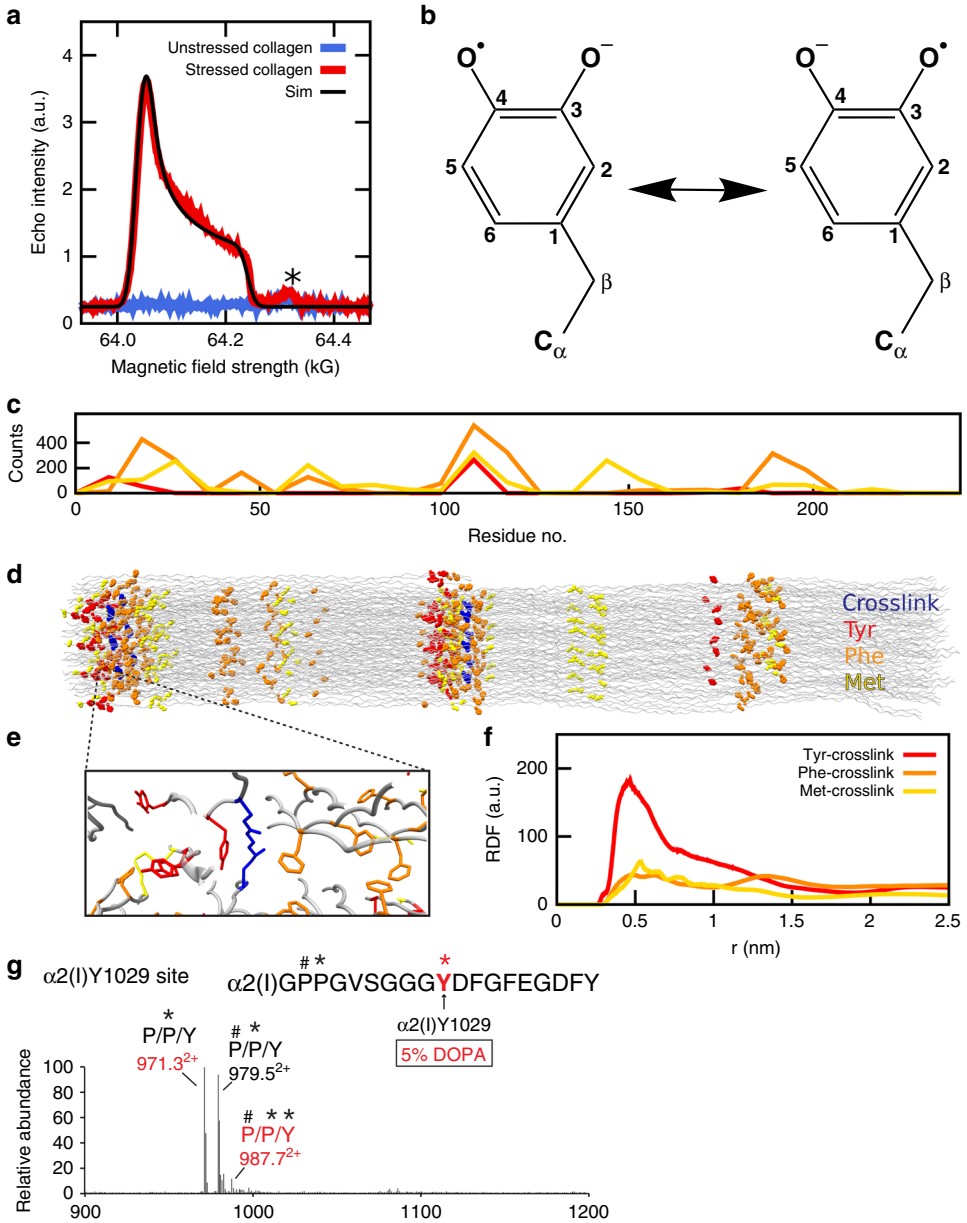

**Fig. 3 DOPA acts as radical sink around crosslinks. a** Echo-detected field-swept EPR powder pattern (G-band frequencies) at 40 K of stressed collagen (red) in comparison to unstressed collagen (blue). The simulated powder pattern (for *g*-tensor values, see Supplementary Table 1) is shown in black. The signal marked by * arises from the quartz capillary. MnO in MgO was used as reference (Supplementary Fig. 3d). **b** DOPA radical anion chemical structure. **c** Average counts of the redox-active residues across vertebrates calculated along the fibril from alignments of representative collagen type I sequences (see "Methods" section). The counts were calculated using a moving average of ten overlapping residues from each chain. **d** Distribution of the redox-active residues in the model of the collagen fibril. **e** Zoom into crosslink region with high density of redox-active residues. **f** Radial distribution function derived from the pulling simulation for redox-active residue-crosslink distances. **g** Mass spectrometry results of a C-terminal peptide of the collagen alpha I chain, showing three posttranslational modification populations ($971.3^{2+}$, $979.5^{2+}$, and $987.7^{2+}$). The peptide sequence of each population is revealed in the MSMS (Supplementary Fig. 3f–h). A DOPA residue is found at Y1029 in peptide ($987.7^{2+}$), which contains 4Hyp, 3Hyp, and 5% DOPA. The *P** indicates 4Hyp; *P#* indicates 3Hyp; *Y** indicates DOPA.

radical anion (Fig. 3b), i.e., the semiquinone, with high spin density on the two oxygens, instead of the previously reported protonated form with spin density on only one oxygen (Supplementary Table 1). The quantitative agreement with the DFT computation of DOPA radical anion in isolation implies that hydrogen bonds of DOPA radical anion to neighboring protein residues or water, which are known to shift the *g*-tensor[30], are either absent or only lead to very minor changes in the signal.

Intriguingly, in the vicinity of crosslinks, the predicted region of bond rupture, collagen I features a strong enrichment in Tyr and Phe (Fig. 3c, d). Tyr and Phe cluster around crosslinks within 0.4–1.5 nm distances, allowing direct or indirect radical transfer (Fig. 3e, f). Both Tyr and Phe can be oxidized to DOPA by oxygen or superoxide, as shown for various proteins including collagen[31]. They play important roles in radical transfer reactions[32], and together (often jointly with Met, Fig. 3c, d) protect proteins against oxidative damage. We were able to identify one DOPA site in the collagen

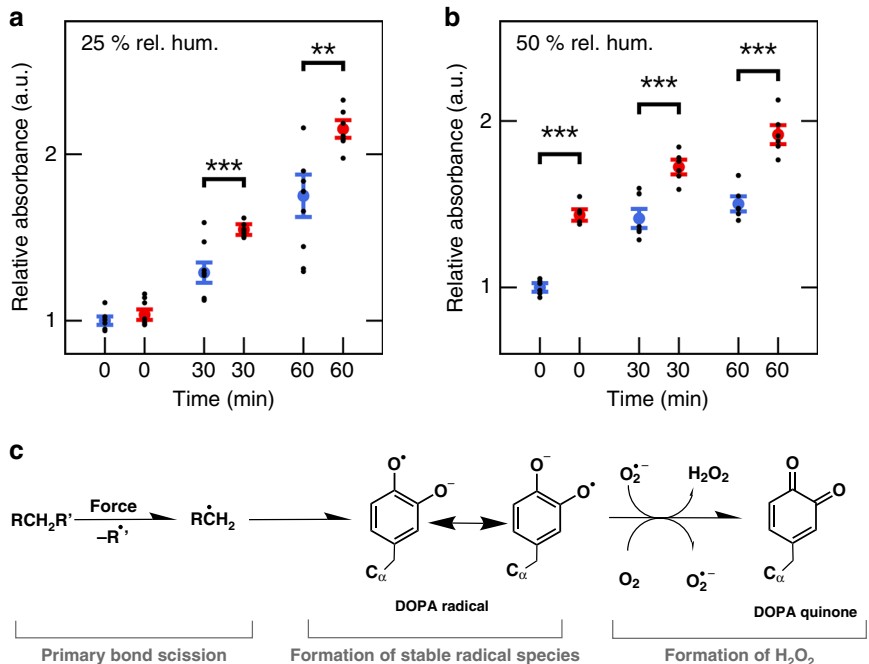

**Fig. 4 Collagen produces hydrogen peroxide. a** Relative absorbance at 595 nm in the FOX assay for different buffer incubation times. Blue: unstressed collagen fascicles, red: fascicles pulled with 15 N, both at 25% relative humidity. Blue and red points show the mean and error bars the standard error of the mean, calculated based on six measurements from two independent sample series (see "Methods" section), shown as black dots. **b** The same plot as in **a** for pulling at 50% relative humidity. Stars in **a** and **b** indicate two-tailed $t$-test $p$-values for samples with equal variance, **$p < 0.01$, ***$p < 0.001$. **c** Scheme proposing a reaction path from primary bond rupture to hydrogen peroxide formation via DOPA radicals.

$\alpha$2-chain at Tyr1029 (at a concentration of ~5%) using mass spectrometry (Fig. 3g, Supplementary Fig. 3f–h). This posttranslational modification is present in unstressed and stressed fascicles.

**Mechanoradicals convert into hydrogen peroxide.** In wet tissue, radicals can react with molecular oxygen in water and form ROS. We used the ferrous oxidation–xylenol orange (FOX) assay to detect hydrogen peroxide in stretched rat tail tendons (see "Methods" section). Figure 4a shows the relative absorption of untreated and pulled collagen fascicles for different buffer incubation times. We detect a significant increase in absorbance at 595 nm after 30 and 60 min of incubation. In pulled tendons, the hydrogen peroxide concentration is ~1 µM higher compared to untreated tendons, as evident from reference measurements (Supplementary Fig. 4a). Performing already the tensile loading of the collagen samples at higher water content, i.e., 50% relative humidity, diminishes the increase in the EPR signal intensity (Supplementary Fig. 4b). However, the FOX measurements on these samples again detect a pronounced increase in $H_2O_2$ concentration by pulling (Fig. 4b). Hence, bond scission still occurs homolytically at this humidity, but produced radicals that readily convert to peroxides. In agreement, we immediately detect peroxides already without incubation in wet collagen samples (Fig. 4a, 0 min, compare Fig. 4b, 0 min).

Hydrogen peroxide can form from DOPA radicals by a process involving superoxide (Fig. 4c). While the EPR measurements described above did not show signatures of the short-lived superoxide, we were able to detect superoxide in the supernatant of stressed tendon collagen I incubated with the spin trap 1-hydroxy-3-methoxycarbonyl-2,2,5,5-tetra-methylpyrrolidine (CMH; Supplementary Fig. 4c, see "Methods" section for details).

**Discussion**
Taken the EPR, MD, and QM results together, our data suggest primary homolytic bond rupture in the crosslink regions of collagen I, followed by radical propagation to proximate DOPA and final conversion to hydrogen peroxide.

Heterolytic bond rupture has been shown to be preferred over homolytic bond rupture if the rupturing bond is directly surrounded by water, which then acts as a nucleophile and initiates hydrolysis as opposed to radical formation[33–35]. Instead, our data show homolytic bond scission to occur to a significant extent in dense collagen tissues even under high humidity, leading to subsequent ROS formation, in close analogy to what has been previously found for hydrated synthetic polymers[3,4]. Collagen water content could potentially define the branching of homolytic versus heterolytic rupture, thereby determining the amount of mechanoradicals at a given external mechanical stress level.

DOPA, arising from the enzymatic- or mechanoradical-mediated oxidation of conserved Tyr or Phe residues around collagen crosslink sites, can act as protective radical sink by taking up the unpaired electron from much less stable primary radicals through radical migration (Fig. 4c). Interestingly, DOPA has been recently observed in other redox-active proteins undergoing electron transfer[28,29]. The observation of DOPA in collagen has also been made previously, with evidence for DOPA-derived crosslinks[36,37]. We propose that such crosslinks could contribute to collagen stiffening and aging. Alternatively, DOPAs can be further oxidized to quinones, see Fig. 4c. Quinones could be potentially recovered by reducing agents, such as ascorbic acid, which is present extracellularly[38]. Quinones also can crosslink with lysines[39] and form metal complexes[40]. Whether these processes play a role in collagen, by stabilizing its structure or by the recovery of DOPA sites, is currently unknown.

Our study proposes collagen mechanoradicals to couple sub-failure mechanical stress to oxidative stress in tendons and supposedly other connective tissues. The biological roles of tension-induced increases in extracellular ROS levels are currently unknown. A very compelling indication for a biological relevance of collagen mechanoradicals is the highly conserved accumulation

of ROS-scavenging residues (Tyr/Phe/Met) around collagen crosslinks (Fig. 3). This enrichment across species suggests force-induced oxidative stress as an evolutionary driving force for collagen I. A second line of support comes from the concentration of DOPA radicals (1–10 μM, from spin count estimate, Fig. 1a, b) and hydrogen peroxide (~μM, Fig. 4a, b). These concentrations appear high enough to impact downstream processes, such as oxidation of membranes, membrane proteins, or others. Hydrogen peroxide stemming from mechanoradicalas can potentially play a role in tissue homeostasis, physical exercise[41], and redox-mediated pathologies, such as pain, inflammation, and arthritis.

## Methods

**Tensile tests and EPR.** Tendons were cut out of Sprague Dawley rat tails from Charles River using a disposable surgery scalpel. Tendon samples were ~2 cm long with cross-sectional areas varying along and among samples between 0.5 and 3 mm², and with weights between 20 and 30 mg. After extraction, the collagen tendons were equilibrated at least 1 h at RT and different humidities. Tensile tests were performed with LEX810 from Diastron covered by a custom-built humidity chamber to control humidity. Tendons were pulled with forces in the range of 5–20 N that corresponds to stresses of ~2–40 MPa depending on the tendon cross-sectional area, 0.5–3.0 mm². The constant force was reached by increasing the extension within ~200 s using creep rates of 0.01 mm/s, and then kept for 1000 s. Cyclic stretching was performed with a creep rate of 0.05 mm/s reaching the maximum force. Then fully relaxing the tendon, kept it without force for 1 s and stretching again. This process is repeated 10, 20, or 50 times.

For cw-EPR measurements at X-band (~9 GHz frequency), the stressed tendons were put into 3 mm EPR tubes and measured at RT (or liquid nitrogen temperature, Supplementary Fig. 3b, c) with a Bruker ESP300 setup with standard parameters (2 mW microwave power, 0.2 mT modulation amplitude, 100 kHz modulation frequency, and 20 scans per sample and load). For the real-time measurement, the tendon was attached inside the EPR cavity and pulled at the same time with a load of 350 g that corresponds to a force of 3.43 N, using the same EPR setup and parameters. To prevent radical transfer, we additionally performed both the mechanical loading and the EPR measurement at 77 K. For mechanical loading, we crushed the sample in a mortar for 5 min in liquid nitrogen. We confirmed that crushing in the mortar produces the same type of radicals by crushing and measuring at RT and comparing the X-band spectra (Supplementary Fig. 3a).

G-band EPR measurements were performed at 40 K with a custom-built spectrometer[42], upgraded to 100 mW microwave power on the transmitter output, using a microwave frequency of 180 GHz and a tendon collagen sample crushed at RT. The absorption signal is recorded directly, compared to cw-EPR, where the first derivative is recorded. Pulse mode detection of the EPR signal with a Hahn echo sequence was used for the G-band measurements. The pulse lengths for the π/2- and π-pulse were 50 ns and 100 ns, respectively. The pulse separation time was 250 ns and experiment repetition time was 10 ms. The magnetic field sweep rate was 1 G/s and 100 signals were averaged per magnetic field step. The linearity of the magnetic field was controlled by a MnO in MgO sample as standard reference (see Supplementary Fig. 3d).

**Collagen fibril modeling.** A full-atom model of collagen fibril from *Rattus norvegicus* was built using an integrative structural modeling approach. In this approach, the overall shape of collagen molecules and their relative packing within collagen fibril was based on the low-resolution fibril structure from fiber diffraction (*R. norvegicus*, PDB code: 3HR2[21]), whereas the local interactions were modeled based on distance restraints derived from high-resolution collagen structures. These restraints were generated based on the collagen model built using THeBuScr program[43], which allows building idealized collagen models based on statistical parameters derived from high-resolution collagen-like peptides structures. First, the atomic model of the collagen triple helix was constructed using Modeller[44] by taking the 3HR2 structure as a template and replacing the template inter-atomic distance restraints with the high-resolution restraints. Then, the model of the full collagen fibril was reconstructed by applying the symmetry information derived from 3HR2 structure to the triple helix model. The fragment spanning one gap and one overlap region[21,22] of a bundle of 37 aligned collagen I triple helices (Fig. 2a) was selected as a representative model. Acetyl and N-methyl groups were attached to the N- and C-termini of truncated polypeptide chains, respectively. Steric clashes between side chains were resolved using relax protocol in Rosetta[45,46]. To reconstruct covalent HLKNL crosslinks between modified lysine residues first the lysine residues were mutated in silico to modified residues corresponding to two halves of a crosslink using Modeller, connected by special bonds and followed by energy minimization using GROMACS[47]. This resulted in the model comprising 12 HLKNL crosslinks (six N-terminal and six C-terminal).

**MD simulations.** Structures of crosslink molecules were built using Maestro[48] and geometry optimized with Gaussian09[49], using the B3LYP functional[50,51]. Restrained electrostatic potential atomic partial charges were calculated using Antechamber[52,53]. Parameters were derived using Acpype[54]. MD simulations were performed using the GROMACS software[47], with the Amber99SB-ildn*[55,56] force field and a time step of 2 fs. For the simulations, the model of the collagen fibril was solvated in a TIP4P[57] water box of 16 × 17.5 × 140 nm³. Then Na⁺ and Cl⁻ ions were added at a concentration of 150 mM. An initial energy minimization was performed using the steepest descent method. Thereafter, 10 ns of NVT followed by 10 ns of NpT equilibration were performed, both with harmonic positional restraints on the protein heavy atoms of 1000 kJ/mol/nm². Subsequently, 50 ns of equilibration with releasing harmonic restraints, followed by 50 ns unrestrained equilibrium MD simulations were carried out. The temperature was kept constant at 300 K by using a velocity rescaling thermostat[58], with a coupling time of 0.1 ps. The pressure was kept constant at 1 bar using isotropic coupling to a Parrinello-Rahman barostat[59], with a coupling time of 2.0 ps. In all simulations, the long-range electrostatic interactions were treated with the particle mesh Ewald method[60]. All heavy atom–hydrogen bonds were constrained using the LINCS procedure[61].

The obtained equilibrated system was subjected to 100 ns constant force pulling simulation using an average force of 1 nN per chain, equally distributed among different triple helices. We allowed the system to equilibrate under load for the first 10 ns, and then monitored forces in bonds[24] for the following 10–100 ns of the simulation. In brief, time-averaged scalar pairwise forces $F_{ij}$ from bond potentials were computed for atom pairs $i$, $j$ of backbone and crosslinks. We also analyzed the differences in the calculated bond forces $F_{ij}$ between simulations under force, and in absence of force and obtained very similar results, i.e., forces in bonds during equilibrium MD without external load are negligible on average. Additional simulations were carried out with the same average force of 1 nN per chain but unequally distributed into the helices, denoted shear loading (Supplementary Fig. 2), to take the effect of structural irregularities of real fibrils into account. To mimic the behavior of the longer collagen fibril, we prevent unwinding of triple helices at the termini that resulted from the truncation of the system. Torque restraints at the termini of the triple helices were applied using the enforced rotation protocol[62] during both equilibration and pulling simulations.

**Sequence analysis.** To obtain the multiple sequence alignments (MSAs) of COL1A1 and COL1A2 sequence families for conservation analysis, 3614 collagen sequences belonging to the collagen orthologous group (KOG3544) were downloaded from the EggNOG database[63]. The sequences were then clustered using CLANS[64]. A cluster of sequences containing COL1A1 and COL1A2, and other closely related collagens was extracted and aligned using MAFFT[65]. From the resulting MSA, a neighbor-joining phylogenetic tree was constructed using JalView and, based on the tree, MSAs comprising COL1A1 (64 sequences) and COL1A2 (59 sequences) were extracted.

To assess if the location of redox-active residues in the vicinity of crosslinks is conserved within the two collagen families, a repetitive bundle (five triple helices) of the collagen fibril was selected and divided into slices spanning ten residues in length along the fibril. The number of the given amino acid type in each slice was calculated by summing up all occurrences of that residue in the sequences within the MSA in the positions of the residues located within the slice. Protein structures were visualized with UCSF Chimera[66].

**QM calculations.** Dimethylamine and major part of HLKNL crosslink without the optional hydroxyl group as representatives for collagen backbone and collagen crosslinks were calculated. First, structures were optimized with constraints on the first and the last heavy atom with Gaussian09[49], using B3LYP[51] and cc-pVDZ basis set[67]. The distance between atoms was increased in steps of 0.05 Å that corresponds to an elongation of every bond by ~5% along an axis. After that, an ab initio multireference CASSCF calculation was performed on the optimized structures using Molcas 8[68] to calculate the bond dissociation energy of the ground state and the first excited state, respectively. The active space contained six orbitals populated by six electrons. EPR g-factors were calculated by unrestricted B3LYP calculations with cc-pVDZ basis set of optimized DOPA radical structures in their protonated and unprotonated forms using ORCA 4.0[69].

**Mass spectrometry.** Rat tail tendon collagens were solubilized from the tissues by acid extraction in 0.5 M acetic acid for 24 h at 4 °C. Collagen α-chains were resolved by 6% SDS–PAGE and stained with Coomassie Blue R-250. Whole tissues were also digested with bacterial collagenase and total collagenase digests were resolved into peptide fractions by C8 reverse-phase HPLC[70]. Collagen degradation products were identified by mass spectrometry and quantified based on band intensities using National Institutes of Health ImageJ software.

Electrospray LC–MS was carried out on in-gel trypsin-digested peptides and collagenase-digested peptides, using an LTQ XL linear quadrupole ion-trap mass spectrometer equipped with in-line Accela 1250 LC and automated sample injection (ThermoFisher Scientific)[71,72]. Proteome Discoverer software (ThermoFisher Scientific) was used for peptide identification. Peptides were also identified manually by calculating the possible MS/MS ions and matching them to

the actual MS/MS spectrum, using Thermo Xcalibur software. Protein sequences used for MS analysis were obtained from the Ensembl genome database. The mass spectrometry data are available in MassIVE repository, ftp://massive.ucsd.edu/MSV000085052/.

**Superoxide detection with spin traps**. CMH was stored as 10 mM stock at −20 °C. For the experiment, the CMH stock solution was diluted to a 600 µL working solution with CMH assay buffer (1× PBS w/o Mg2+/Ca2+, 5 µM diethyldithiocarbamate, 25 µM deferoxamine). The working solution was prepared freshly for each set of experiments and kept on ice. For the incubation, washed and tried fascicles (preparation described in "Tensile test and EPR") were placed in a 0.5 mL reaction tube, overlayed with 250 µL CMH working solution, and incubated for 31 min at RT. At time points 1 min, 11 min, 21 min, and 31 min, 50 µL supernatant were aspirated directly into disposable micropipettes, inserted in quartz EPR tube. In independent experiments, unstressed, pulled with 15 N and crushed fascicles were measured. For the samples crushed in the mortar, no sample was taken at time point 1 min. The CMH supernatant was measured with the same cw X-band EPR setup and parameters described in "Tensile test and EPR", with five scans each at an attenuation of 54 or 60 dB. The sum of spectra was normalized to the dry weight of the respective fascicle's mass.

**Hydrogen peroxide detection**. Multiple rat tail tendon samples were extracted the same way as described in section "Tensile test and EPR". After extraction, the tendons were rinsed, vortexed, and centrifuged for 2 min with 2000 rpm in PBS buffer without calcium and magnesium chloride, and after that dried for 2 h at RT. The tendon samples were separated in two groups of equal mass, each tendon from the first group was pulled for 200 s with 15 N (reached with a creep rate of 0.05 mm/s) as described in section "Tensile test and EPR". Tendons in the second group were kept untreated for reference. The procedure was done at 25% and 50% relative humidities.

After treatment, the tendons were put into 2 ml Eppendorf tubes together with 1.7 ml PBS, without calcium chloride and magnesium chloride. After 0 min, 30 min, and 60 min, 100 µl supernatant was taken from both, treated and untreated samples, and pipetted in Corning 96 Flat Bottom Transparent Polystyrol well plates. To measure the peroxide concentration, 100 µl working reagent of Pierce™ Quantitative Peroxide Assay Kit from ThermoScientific was added to each well and incubated for 1000 s in the dark at RT. The oxidation of ferrous to ferric ion in the presence of xylenol orange was used to detect peroxide (FOX). The absorption of xylenol orange was measured with a TECAN infinite M200 Pro at 595 nm. Per sample and every point in time, three values from different wells were collected.

**Reporting summary**. Further information on research design is available in the Nature Research Reporting Summary linked to this article.

## Data availability
Data supporting the findings of this manuscript are available from the corresponding author upon reasonable request. A reporting summary for this article is available as Supplementary Information file. The source data underlying Figs. 1a–c, 2a, c, 3a, c, f and 4a, b, and Supplementary Figs. 1b–d, f, 2b, d, e, 3a–d, f–h and 4a–c are provided as Source data file. The mass spectrometry data that support the findings of this study are available in MassIVE repository, ftp://massive.ucsd.edu/MSV000085052/.

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

## Acknowledgements

We are grateful for financial support by the Klaus Tschira Foundation, the Volkswagen Foundation, the Excellence Cluster Cellnetworks, and BIOMS of Heidelberg University. The authors acknowledge support by the state of Baden-Württemberg through bwHPC and the German Research Foundation (DFG) through grant INST 35/1134-1 FUGG. We thank the Center of Breath Research of Saarland University Medical Center for donating the rat tails, mechanical workshop of Saarland University Medical Center for technical support, and Martin Müller of Kiel University for the loan of the LEX810. We thank Marilyn Archer for technical assistance with the collagen extraction and SDS–PAGE.

## Author contributions

C.Z. performed collagen tensile tests, X-band EPR experiments, QM calculations, hydrogen peroxide detection, and together with M.K. spin trap experiment. A.O.-K. performed collagen fibril modeling, MD simulations, and sequence analysis. B.R., M.K., D.M., and A.O.-K. supported tensile tests. D.M.H. performed mass spectrometry and analysed the MS data. C.Z. and M.K. supported mass spectrometry. B.R., U.B., and T.P.D. supported hydrogen peroxide detection. C.Z. and V.D. performed G-band EPR experiments, and T.P. and M.B. supported G-band EPR experiments and their interpretation. C.D. supervised QM calculations. R.K. supervised experiments. F.G. supervised the project. F.G., C.Z., and A.O.-K. wrote the manuscript. All authors commented on the manuscript and contributed to it.

## Competing interests

The authors declare no competing interests.
