## [Peer Review File · Nature Communications]

Reviewers' Comments:

Reviewer #1:

Remarks to the Author:

The paper addresses the induction of mechanoradicals in tensed tendon collagen as a new source of oxidative stress. The paper touches on a novel aspect of tendon working and has potential to be of interest to the broader readership of Nature Communications, but there are a number of points that should be addressed first.

The strong points:

- Although it has been known for a very long time that mechanical stress can induce radicals in different materials, including polymers, this is the first time that it has been shown to happen in tendons.
- The study investigates and details the nature of the formed radical, showing that an initial peroxy radical very fast converts into a stable radical, that is found to be a deprotonated DOPA radical
- The relevant experimental and theoretical techniques are used.

The weaker point:

- Although the radical-formation effect is clearly observed, it is unclear how big this effect is compared to the heterolytic bond rupture, which may be happening in parallel and may be the more important feature.

Questions to be addressed:

- Supplementary table 1: (i) experimental error should be given on g values; (ii) it would be better to give the hyperfine values in MHz, not Gauss, since the latter unit depends on the g values; (iii) why are only isotropic hyperfine values computed? The spectra are solid-state EPR spectra, so the full hyperfine tensor is relevant here and may alter completely the statement that the proton hyperfine coupling is small. Please add the full computed hyperfine tensors.
- P6 + suppl. Info Figure 3: experiments at 77K. The authors demonstrate that at room temperature, crushing and stretching leads to the same radical, but this does not prove that the radical observed by crushing at 77K is indeed an intermediate, non-stable radical formed at room temperature upon mechanical stress. It can also be a specific radical induced because a frozen (and hence rigid) tendon is crushed. This should be commented on.
- Suppl. Figure 3c + suppl. Table 1: I am very much puzzled about the EPR spectrum of the methylene radical and the related g tensor that is mentioned in table 1. If I understand the text correctly, this spectrum is obtained by subtracting two EPR spectra recorded with different microwave power, so exploiting the different saturation behavior of the two radicals. This is done for experiments at 77 K, so this is a solid-state EPR spectrum. I cannot understand how you can get a simulated low-temperature spectrum with three sharp lines, with only a g tensor and no hyperfine values. Please comment on this.
- Why were no spin trapping experiments tried to scavenge potential ROS to parallel the FOX assay? Furthermore, although the FOX assay seems to indicate that there is ROS formation associated with the tendon stressing, this is not a proof that the ROS are formed through the observed mechanoradicals. There may be other mechanisms responsible for this. The authors should address this more carefully and should elaborate more on the potential biological significance of the radical-formation effect.
- Small comment: in abstract: quantum-chemical calculations instead of quantum calculations

Reviewer #2:

Remarks to the Author:

This report combines EPR measurements and molecular dynamics calculations of stretched collagen fibrils. The results are interpreted as the breaking of cross-links in collagen, leading to the formation of radicals visible in EPR.

All the conclusions of the Paper rely on the intensity change of the EPR signal with load. The

accompanying molecular dynamics calculations only make sense in combination with this EPR signal, which is known since 1969 (ref 20) to occur even in unloaded tendon. Its intensity was reported to depend on temperature and hydration of the tendon.

For this reason, the interpretation of the intensity change of the EPR signal in terms of bond breaking is not convincing. Indeed, the hydration level of tendon is known to change with applied load. Based on the 1969 report, it is therefore a possibility that the intensity change of the EPR signal is just due to this trivial effect and is not at all related to bond breaking.

One simple test that one could do is to perform cyclic stretching experiments. After stress release, water would flow back into the tendon and one would expect a reversible intensity change of the EPR signal if hydration is the important parameter. Radicals generated by the breaking of cross-links would not lead to reversible intensity changes, as the bonds are not expected to reform immediately.

Given the fact that all conclusions of the present study depend in a fundamental way on the interpretation of the EPR signal, it is essential to ascertain that cross-links are actually broken under the conditions of the experiment.

Cross-link densities in collagen are now almost routinely measured by mass spectrometry. If measurements of this type could be done, the conclusions could be made definitive.

Without the additional experiments mentioned above (or other approaches that confirm the breaking of cross-links independently of the EPR signal), the conclusions of the paper remain unconvincing.

Reviewer #3:

Remarks to the Author:

Zapp and coworkers present an interesting report that combines experimental and simulation data to put forward the idea that collagen produces radicals under mechanical stress, and hypothesize that aromatic residues in collagen could act as a sponge for these radicals. While the manuscript is well written, the presentation is often too concise and difficult to understand without having to read the text several times. I have the following comments/questions:

The experimental evidence clearly supports the hypothesis that radicals are formed by mechanical stress and induce the formation of tyrosyl radicals. However, more evidence is needed to support the hypothesis that radicals are formed due to bond scission in the vicinity of crosslinking regions. The MD simulations suggest that crosslinking regions may be under larger stresses under load, yet this could potentially be an artifact of the way the simulation model was constructed. The authors should find ways to show that this is not the case, and find a way to experimentally validate the MD simulation prediction.

In the MD simulation methods it is not clear for how long was the system run under tension. Was the FDA analysis carried out after the system had equilibrated under load? Did the authors conduct FDA analysis on the collagen fibril in the absence of tension? Is there an intrinsic stress at the crosslink regions in the absence of tension, and if so, is there a significant increase in this region after tension is applied?

The authors suggest a "new role of collagen which is to convert mechanical into oxidative stress in connective tissues." Are they proposing that this plays a role in some biological function? Or is it just an unwanted consequence of mechanical stress? The authors should be more clear about this statement and provide further support/evidence if they want to suggest a new biological role for this process.

The authors suggest "collagen I to have evolved as a radical sponge against mechano-oxidative damage". If this is the case, what would be a recovery mechanism for this tissue? Experiments seem to show that the radicals formed are stable for long periods, would the lifetime of these radicals change in vivo? Would similar radicals be produced in the neighboring tissues? If so, would collagen also act as a sponge for those radicals?

The authors should be more clear about what EPR measurements were conducted while the fascicles were under tension, and which measurements were conducted after the fascicles were extended for some period of time and then removed from the tensile testing apparatus (this may be obvious to those familiar with the experimental setup, but it is not clear from the text).

Panels b and c in Figure 1 should have legends that show more clearly the difference between the curves.

Reviewer #1:

The paper addresses the induction of mechanoradicals in tensed tendon collagen as a new source of oxidative stress. The paper touches on a novel aspect of tendon working and has potential to be of interest to the broader readership of Nature Communications, but there are a number of points that should be addressed first.

The strong points:

- Although it has been known for a very long time that mechanical stress can induce radicals in different materials, including polymers, this is the first time that it has been shown to happen in tendons.
- The study investigates and details the nature of the formed radical, showing that an initial peroxy radical very fast converts into a stable radical, that is found to be a deprotonated DOPA radical
- The relevant experimental and theoretical techniques are used.

The weaker point:

- Although the radical-formation effect is clearly observed, it is unclear how big this effect is compared to the heterolytic bond rupture, which may be happening in parallel and may be the more important feature.

Questions to be addressed:

- *Supplementary table 1: (i) experimental error should be given on g values; (ii) it would be better to give the hyperfine values in MHz, not Gauss, since the latter unit depends on the g values; (iii) why are only isotropic hyperfine values computed? The spectra are solid-state EPR spectra, so the full hyperfine tensor is relevant here and may alter completely the statement that the proton hyperfine coupling is small. Please add the full computed hyperfine tensors.*

We adopted the units as suggested and added the diagonal hyperfine tensor to the SI (new Suppl. Table 2).

- *P6 + suppl. Info Figure 3: experiments at 77K. The authors demonstrate that at room temperature, crushing and stretching leads to the same radical, but this does not prove that the radical observed by crushing at 77K is indeed an intermediate, non-stable radical formed at room temperature upon mechanical stress. It can also be a specific radical induced because a frozen (and hence rigid) tendon is crushed. This should be commented on.*

We agree. Our key point is that radicals from bond scission convert into DOPA radicals, and a peroxyradical can but must not be an intermediate and can instead be specific to the 77K case. We discuss this more carefully now on page 6 and modified Fig. 4 accordingly to clearly state that the peroxyradical can be but must not be an intermediate at room temperature.

- *Suppl. Figure 3c + suppl. Table 1: I am very much puzzled about the EPR spectrum of the methylene radical and the related g tensor that is mentioned in table 1. If I understand the text correctly, this spectrum is obtained by subtracting two EPR spectra recorded with different microwave power, so exploiting the different saturation behavior of the two radicals. This is done for experiments at 77 K, so this is a solid-state EPR spectrum. I cannot understand how you can get a simulated low-temperature spectrum with three sharp lines, with only a g tensor and no hyperfine values. Please comment on this.*

We apologize for not adding the hyperfine values to the table in the first place, and now have added the values for the methylene, see Suppl. Tables 1/2.

- *Why were no spin trapping experiments tried to scavenge potential ROS to parallel the FOX assay? Furthermore, although the FOX assay seems to indicate that there is ROS formation associated with the tendon stressing, this is not a proof that the ROS are formed through the observed mechanoradicals. There may be other mechanisms responsible for this. The authors should address this more carefully and should elaborate more on the potential biological significance of the radical-formation effect.*

We followed the suggestion of the reviewer and performed spin trap experiments (Supplementary Figure 4c). We successfully detected ROS, more specifically superoxide, using the 1-Hydroxy-3-methoxycarbonyl-2,2,5,5-tetramethylpyrrolidine (CMH), and observed higher concentrations for stressed versus unstressed tendon.

The superoxide detection was particularly challenged by the overall low concentration of DOPA radicals and hydrogen peroxide (microM regime) as well as competing ROS forming processes and radical quenching. Since incubation with buffer and CMH requires different biological samples for comparison between stressed and unstressed tissue (in contrast to the EPR experiments of Fig 1 where we directly compared the very same sample), the variance across experiments is quite high. The diffusion of the trap into the tendon and of the radical adduct back into the supernatant poses an additional challenge, and CMH appeared the best choice in this respect. As a consequence, the difference was only statistically significant when comparing crushed to unstressed (but not pulled to unstressed) samples (p-value 0.3×10^{-4} , t-test, two-tailed, unpaired). Overall, we conclude that DOPA radicals convert to H₂O₂ through superoxide, very nicely substantiating our mechanism outlined in Fig. 4c.

We strongly believe that the radical and subsequent H₂O₂ formation is biologically relevant (see also our reply to Rev #3, point 3). In short, a

very compelling indication is the highly conserved accumulation of ROS-scavenging residues (Tyr/Phe/Met) around collagen crosslinks (Fig. 3). A second line of support comes from the concentrations of DOPA radicals (1-10 microM, from spin count of EPR) and of H₂O₂ (~microM). These concentrations are high enough to impact downstream processes such as oxidation of membranes, membrane proteins, or others, thereby affecting pain sensation, inflammation processes or alike. We discuss these points and potential biological implications in more depth in the Discussion.

- *Small comment: in abstract: quantum-chemical calculations instead of quantum calculations*

We added quantum-chemical calculations to the abstract.

Reviewer #2:

This report combines EPR measurements and molecular dynamics calculations of stretched collagen fibrils. The results are interpreted as the breaking of cross-links in collagen, leading to the formation of radicals visible in EPR.

All the conclusions of the Paper rely on the intensity change of the EPR signal with load. The accompanying molecular dynamics calculations only make sense in combination with this EPR signal, which is known since 1969 (ref 20) to occur even in unloaded tendon. Its intensity was reported to depend on temperature and hydration of the tendon.

For this reason, the interpretation of the intensity change of the EPR signal in terms of bond breaking is not convincing. Indeed, the hydration level of tendon is known to change with applied load. Based on the 1969 report, it is therefore a possibility that the intensity change of the EPR signal is just due to this trivial effect and is not at all related to bond breaking.

One simple test that one could do is to perform cyclic stretching experiments. After stress release, water would flow back into the tendon and one would expect a reversible intensity change of the EPR signal if hydration is the important parameter. Radicals generated by the breaking of cross-links would not lead to reversible intensity changes, as the bonds are not expected to reform immediately.

We followed this suggestion and performed cyclic stretching experiments instead of the constant loading scheme we focused on so far. Results are included in Suppl. Fig. 1 d. We recovered the canonical DOPA signal at intensities similar to those seen in our constant force experiments. This supports irreversible bond rupture to give rise to the signal.

We note, however, that water also leads to a vanishing EPR signal (Suppl Fig 4) due to the reaction to H₂O₂ (Fig. 4). An experiment not possible to do is to monitor the EPR signal intensity on the fly during the cyclic stretching, similar to our live measurements under constant load within the EPR shown in Fig 1c. Such an experiment could give more direct evidence of radicals partly vanish due to the inflow of water during stretch release.

More importantly, we now show in independent experiments directly bond scission in tendon collagen (see next paragraph).

Given the fact that all conclusions of the present study depend in a fundamental way on the interpretation of the EPR signal, it is essential to ascertain that cross-links are actually broken under the conditions of the experiment.

Cross-link densities in collagen are now almost routinely be measured by mass spectrometry. If measurements of this type could be done, the conclusions could be made definitive.

Without the additional experiments mentioned above (or other approaches that confirm the breaking of cross-links independently of the EPR signal), the conclusions of the paper remain unconvincing.

Upon your suggestion, we teamed up with the group of David Hudson, University of Washington. SDS-PAGE and densitometry analysis showed a 'tail' of collagen I fragments (identified with mass spectrometry) smaller than the alpha 1 and 2 chains of collagen I (new Suppl. Fig. 1d,e) in both stressed and unstressed samples, with a higher proportion of smaller molecular weight fragments in stressed samples. This supports the mechanism suggested in this paper that radicals originate from mechanical load, constant or cyclic, through bond scission in the collagen molecules.

The disadvantage of the SDS-PAGE is the fact, that while we see an integrated signal for all radicals created in the EPR, the fragments of the fibers smear out on the gel becoming more difficult to detect and quantify. Thus, we think that quantification of bond cleavage is still very challenging to accomplish - especially because we consider the scission to be a rare event compared to the total number of molecular chains. Only under extreme conditions, such as the crushing of collagen, such events become easier to quantify.

We also note that our simulations predict scission next to crosslinks to be preferred over scission directly within the crosslinks (see our answer to Rev #3 in this regard and new Ref 25), and the SDS-PAGE is suited to pick up such scissions within the protein backbone. We still will follow up the suggested crosslink density analysis by MS to test for scissions in the

crosslinks, which we, however, imagine again very challenging given the expected low relative changes.

By means of MS, however, we successfully detected DOPA at one specific Tyr site in the collagen I alpha2 chain (new Figure 4g and Suppl. Fig. 3f-h), validating the interpretation of our EPR G-band signal.

Reviewer #3:

Zapp and coworkers present an interesting report that combines experimental and simulation data to put forward the idea that collagen produces radicals under mechanical stress, and hypothesize that aromatic residues in collagen could act as a sponge for these radicals. While the manuscript is well written, the presentation is often too concise and difficult to understand without having to read the text several times. I have the following comments/questions:

1. The experimental evidence clearly supports the hypothesis that radicals are formed by mechanical stress and induce the formation of tyrosyl radicals. However, more evidence is needed to support the hypothesis that radicals are formed due to bond scission in the vicinity of crosslinking regions. The MD simulations suggest that crosslinking regions may be under larger stresses under load, yet this could potentially be an artifact of the way the simulation model was constructed. The authors should find ways to show that this is not the case, and find a way to experimentally validate the MD simulation prediction.

Let us start by emphasizing that the key point of our study is the formation of mechanoradicals and subsequent ROS formation in collagen. Where these radicals form is a very interesting mechanistic question but overall of minor relevance. Beyond our MD simulations, we have three lines of evidence where rupture occurs:

1. Our new experimental data directly verifies bond scission upon stretching using SDS-PAGE (Suppl. Fig. 1d,e). It suggests that bond cleavage does not happen at one given specific site but instead leads to a broad distribution of fragments, as expected. If crosslink regions are particularly prone to rupture cannot be inferred from that data.

2. We now have included mass spec data (Fig. 3g and Suppl. Fig. 3f-h) that identifies Tyr1018 as a DOPA site. This site is located very close to a crosslink, indirectly suggesting radical transfer from a nearby cleavage site to DOPA.

3. Other Tyr/Phe are clustered around crosslinks, further corroborating, albeit again indirectly, the prevalence of mechanoradical formation in this region.

We finally note that during the revision process, a methodological MD paper of our group was accepted in JCTC which directly simulates bond rupture in collagen and again finds, as suggested by the high bonded forces, rupture to preferentially - but not exclusively - occur in the vicinity of crosslinks. We now cite this study on page 5. For convenience, we here include a visualization of bond rupture propensities in the fibril under force:

Figure A from <https://doi.org/10.1021/acs.jctc.9b00786>: Location of rupture events in the collagen system are concentrated near the crosslinks (depicted in black). Atoms within broken bonds are enlarged with a sphere radius that scales with the respective frequency of the cleavages.

2. In the MD simulation methods it is not clear for how long was the system run under tension. Was the FDA analysis carried out after the system had equilibrated under load? Did the authors conduct FDA analysis on the collagen fibril in the absence of tension? Is there an intrinsic stress at the crosslink regions in the absence of tension, and if so, is there a significant increase in this region after tension is applied?

The system was simulated under load for 100 ns. After an equilibration of the external force for 10ns, FDA was carried out for the following 10-100 ns of the simulation. The missing information was added to the Methods section.

We performed FDA analysis for the system under load, i.e. without subtracting the bonded forces from equilibrium simulations as reference, since this appears the more relevant measure for the propensity to undergo bond scission in the system under load (Fig2 in the manuscript).

Nevertheless, we also performed the FDA for the system in the absence of load and subtracted the bond forces to analyze the increase in forces solely due to the externally applied load itself. We recovered the same tendencies (see Figure below) so bond forces in equilibrium simulations of collagen, also in crosslinks, are comparably small. We discuss this point in the Methods section.

Figure B: As in main text Figure 2, but instead of absolute bond forces as present in MD simulations under an external constant force, differences between bond forces in MD simulations with and without an external force are shown.

3. The authors suggest a “new role of collagen which is to convert mechanical into oxidative stress in connective tissues.” Are they proposing that this plays a role in some biological function? Or is it just an unwanted consequence of mechanical stress? The authors should be more clear about this statement and provide further support/evidence if they want to suggest a new biological role for this process.

Our current study shows the generation of oxidative stress from mechanical stress. How this oxidative stress influences biological process (for good or bad) is not part of the current study. However, we believe we have strong indications for biological relevance:

1. We now have estimated the number of spins in a pulled tendon piece of 40 mm³ by comparison of the EPR intensity to a reference of 2,2-diphenyl-1-picrylhydrazyl (DPPH) at known concentration within the same volume of

40 mm³ (defined by soaking a piece of paper of this approximate volume). We obtained ~8 microM of spin concentration on the collagen fibril under 15N of force. This is qualitatively in line with the ~microM concentration H₂O₂ in the supernatant (Fig. 4a/b and Suppl. Fig. 4a) and a concentration with potential impact on the extracellular redox stress levels.

2. New Fig. 3g gives direct evidence from mass spec for DOPA in collagen. DOPA has been previously shown to be involved in radical migration in ribonucleotide reductase in mitochondria (Refs 28/29), i.e. in a highly oxidative environment. Collagen containing DOPA, by analogy, likely performs a similar function of a radical scavenger. In addition, Tyr/Phe/Met enrichment is a hallmark of mitochondrial proteins and is similarly present in clusters of the 3D collagen fibril. Taken together, oxidative stress appears to have shaped the collagen sequence and post-translational modification to DOPA, underlining the biological importance of mechano-redox coupling.

Both points above are now part of the extended Discussion section.

4. The authors suggest “collagen I to have evolved as a radical sponge against mechano-oxidative damage”. If this is the case, what would be a recovery mechanism for this tissue? Experiments seem to show that the radicals formed are stable for long periods, would the lifetime of these radicals change in vivo? Would similar radicals be produced in the neighboring tissues? If so, would collagen also act as a sponge for those radicals?

These are very interesting questions. DOPA radicals can combine, thereby forming DOPA crosslinks (McDowell et al. 1999, 10.1074/jbc.274.29.20293; Kato et al., 1993, 10.1111/j.1751-1097.1994.tb05045.x) and potentially leading to stiffening during collagen ageing. Alternatively, they can be further oxidized to quinones, see Fig. 4. Quinones could be potentially recovered by reducing agents such as ascorbic acid, which is present extra-cellularly (e.g. Carr et al Am J Clin Nutr 2013;97:800–7.). Quinones also can crosslink with cysteines (which however are absent in assembled collagen) and lysines (Li et al, Free Rad. Biol. Med. 97(2016)148–157). If these processes play a role in the recovery of DOPA sites is currently unknown.

The radical life time is long (over hours) in vitro (see e.g. Fig. 1c) but will be limited in vivo by the presence of water. In vitro, we do not observe radicals but only the subsequent H₂O₂ at high humidity (close to tendon conditions in vivo) within the minutes time scale of the experiment (Fig 4b and Suppl Fig. 4b). This is a likely scenario also for the in vivo situation. In this light, radical transfer from other tissues is unlikely but restricted to nanometer-scale radical migration mechanisms within collagen where water/oxygen is not in direct proximity. However, other tissues might show

similar mechanoradical formation mechanisms and might have evolved radical sponge proteins (e.g. other collagen types) according

We now discuss both points in the extended Discussion section in more depth.

5. The authors should be more clear about what EPR measurements were conducted while the fascicles were under tension, and which measurements were conducted after the fascicles were extended for some period of time and then removed from the tensile testing apparatus (this may be obvious to those familiar with the experimental setup, but it is not clear from the text).

Results shown in Fig. 1a,b originate from experiments where stretching was followed by the EPR measurement. Fig. 1c shows results from an EPR measurement of a fascicle under force during the measurement.

Panels b and c in Figure 1 should have legends that show more clearly the difference between the curves.

We changed the legends of the figure to improve the understandability (see clarification above).

Reviewers' Comments:

Reviewer #2:

Remarks to the Author:

The authors went a long way to address the reviewer question, adding new experiments. The conclusions are now much better supported the data and the discussion keeps a good balance between what is known and where the ideas are more speculative. I think this is now a very valuable addition to our understanding of collagen mechanics.

Reviewer #3:

Remarks to the Author:

The authors have successfully addressed all my previous concerns with the manuscript.